# Seroprevalence and risk factors associated with exposure to *Leishmania infantum* in dogs, in an endemic Mediterranean region

**Pasquale Rombolà**[1]*, **Giulia Barlozzari**[1], **Andrea Carvelli**[1], **Manuela Scarpulla**[1], **Francesca Iacoponi**[2], **Gladia Macrì**[1]

**1** Istituto Zooprofilattico Sperimentale del Lazio e della Toscana '*M. Aleandri*', Rome, Italy, **2** Istituto Superiore di Sanità, Department of Food Safety, Nutrition and Veterinary Public Health, Rome, Italy

* pasquale.rombola@izslt.it.

**Data Availability Statement:** All relevant data are within the manuscript and its Supporting Information files.

## Abstract

Canine leishmaniasis (CanL) is a zoonotic parasitic disease caused by *Leishmania infantum* in the Mediterranean area and transmitted by phlebotomine sand fly vectors. The domestic dog is the main reservoir host. The aim of this study was to assess the influence of different individual, environmental and spatial risk factors on the dog exposure to *L. infantum* and to estimate the seroprevalence among owned and kennel dogs, in the Lazio region (central Italy), where canine leishmaniasis is endemic. In the period 2010–2014, 13,292 sera from kennel and owned dogs were collected by official and private veterinarians. The presence of anti-*Leishmania* IgG was analysed by indirect fluorescent antibody test (IFAT), using a 1:80 titre cut-off. At the univariable analysis, CanL seropositivity was associated with sex, size, breed, coat length, living with other dogs and forest/semi-natural land cover. At the multivariable analysis, age, ownership and attitude were confirmed as risk factors, being more than 2 years old, owned, and hunting dogs at higher risk. Being a Maremma sheepdog was a protective factor. A true overall seroprevalence of 6.7% (95% CI: 6.2–7.2) was estimated in the whole population while 7.3% (95% CI: 6.8–7.8) was estimated in kennel dogs and 74.3% (95% CI: 70.8–77.6) in owned dogs. The role of kennels as a key component for CanL active and passive surveillance was also highlighted. This study confirmed the endemicity of CanL in the Lazio region and focused some factors that can influence the seropositivity of dogs in a Mediterranean region.

## Introduction

The leishmaniases are a group of zooanthroponotic diseases transmitted among vertebrate hosts by infected females of phlebotomine sand flies (Diptera, Psychodidae). Etiological agents are intracellular protozoa belonging to the genus *Leishmania* (Kinetoplastida, Trypanosomatidae) [1].

The worldwide distribution of leishmaniases is concentrated in tropical and temperate regions in 98 countries, mainly with a poor Human Development Index. In these countries, leishmaniases are considered endemic with an estimated 12 million cases and a number of

**Funding:** The authors received no specific funding for this work.

**Competing interests:** The authors have declared that no competing interests exist.

1.5–2 million new cases occurring annually in a population of 350 million people at risk of infection [2, 3].

In Europe, *Leishmania infantum* is the causative agent of canine leishmaniasis (CanL) and dogs are the main reservoir of the zoonotic type of both visceral and cutaneous leishmaniasis [2]. The role of cats has long been discussed and several studies indicate the infected cats as a domestic additional reservoir host for human infection [4, 5]. The estimated annual incidence of visceral leishmaniasis (VL) in the WHO European Region varies between 1100 and 1900 cases [6]. Italy is among the most affected countries, together with Georgia, Spain, Albania, Turkey, Tajikistan and Azerbaijan, showing an estimated annual VL incidence from 160 to 240. In Italy, the most relevant endemic foci are in southern-central regions such as Tuscany, Sicily, Campania and Sardinia [6]. Notwithstanding, Italy provided the first evidence in Europe of the emergence and northward spread of VL as probable effect of climate change [7].

In dogs, *L. infantum* causes a spectrum of clinical signs that varies greatly from asymptomatic/mild to a very severe disease. Several factors including the immunological status and the genetic background of the host can influence the outcome of CanL [8]. Infected dogs, whether clinically healthy or sick, can be infectious for phlebotomine sand flies that may transmit the infection to other hosts [9].

Serological surveys are a useful tool to quantify the spread of pathogens within a specific area and represent the starting point to develop proper and risk-based approach in surveillance activities. Indirect fluorescent antibody test (IFAT) and enzyme-linked immunosorbent assay (ELISA) are the most commonly used serological methods. In particular, the in-house IFAT prepared with whole body parasite as antigen, following the World Organisation for Animal Health (OIE) procedure is considered the "gold standard" for serological diagnosis [1, 10]. It is suitable for epidemiological studies, individual diagnosis and in treatment follow-up [10]. However, IFAT interpretation is operator-dependent and it requires trained and experienced personnel especially in case of titres around the cut-off. ELISA tests showed variable diagnostic performance depending on the type of antigen used. In particular, ELISA based on soluble promastigote and amastigote antigens had high sensitivities in both symptomatic and asymptomatic dogs, while rK39 ELISA seem to be less sensitive in detecting asymptomatic cases [11]. It is noteworthy to mention that low-medium IFAT titres/optical density values are not necessarily consistent with infection but can be rather indicative of exposure to *L. infantum*, as well as some infected individuals (positive by PCR, cytology/culture) can show low-medium serological titres/optical density values. On the contrary, high serological titres, defined as a 3–4 fold elevation above the cut off level of a reference laboratory or a seroconversion, are strongly indicative of a diagnosis of CanL [8, 12].

Previous serosurveys on CanL carried out in Mediterranean countries reported prevalence ranging from 2% to 30%, whereas 50% or higher values were reported from hyperendemic foci where the majority of population can be infected [13–15]. In Italy, captured stray dogs are housed in kennel fenced outdoor areas with high animal density, where they can be frequently exposed to the bites of vectors. These conditions together with the lack of a regular anti-vectorial prophylaxis represent a potential risk factor for the spread of CanL. Therefore, kennel dogs can act as sentinels for the circulation of several infectious agents, including *L. infantum*, in a specific area [16, 17].

The aim of this study was to assess the influence of different individual, environmental and spatial risk factors on the dog exposure to *L. infantum* as well as to estimate the seroprevalence among owned and kennel dogs in the Lazio region (central Italy), where canine leishmaniasis is endemic.

## Materials and methods

### Study area and canine population

The study was carried out in the Lazio region, central Italy (Fig 1). The area is located on the Tyrrhenian coast. Climate and vegetation are Mediterranean with mild, wet winters and hot, dry summers and maximum rainfall recorded in autumn and spring. This is a highly heterogeneous area regarding altitude, weather conditions, land cover/use and human population density. The region is mainly hilly with a mean altitude of about 400 meters above sea level, ranging from the plain to the mountain.

In Italy, the canine population is composed by almost 12 million of owned dogs [18] and estimated 160,000 stray dogs [19]. Since 1991, every owned dog and caught stray dogs must be identified by a microchip (tattoo before 2005) and registered in a Dog Registry (Legge 281/1991) with data on the animal and owner (i.e. sex, age, breed, address, owner, transfer, date of birth/death, etc.). During the implementation of this study, in the Lazio region, more than 500,000 dogs were registered in the Dog Registry, whose completeness was around 75% as assessed in previous studies [20, 21]. A point map (Fig 1) of the kennel and dog owners' address was made by a geocoding procedure (QGis 3.10.2, Plugin Pelias Geocoding) [22, 23].

**Sampling and data.** Samples were collected from January 2010 to December 2014. In the Lazio region, in accordance with a Regional Law, all dogs entering in kennels are compulsory

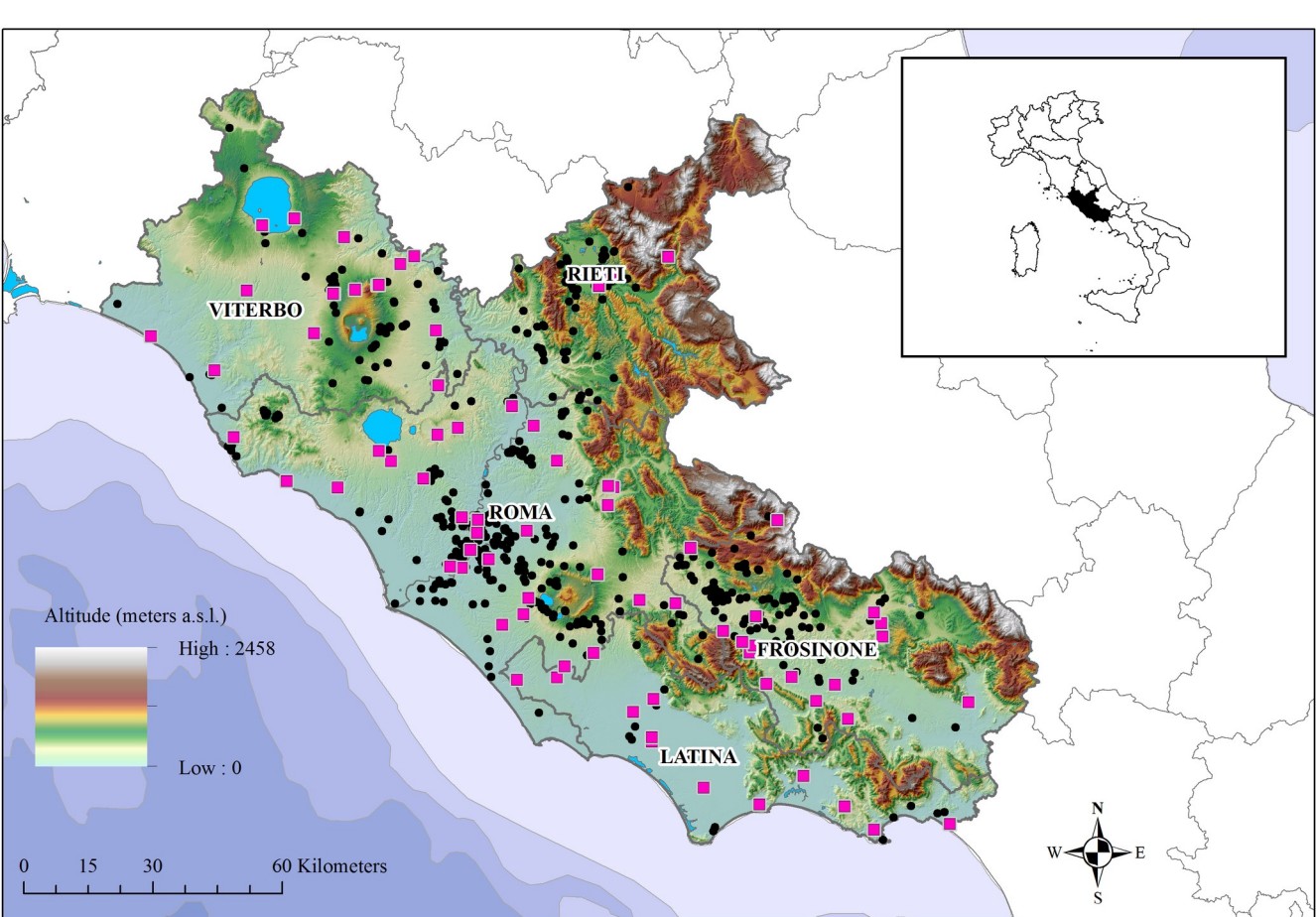

**Fig 1. Location of kennels (pink squares) and owned dogs (black points) tested for *L.infantum*, in 2010–2014, in the Lazio region.**

tested for the presence of antibodies against *L. infantum* by sending blood samples to laboratories accredited under ISO/IEC 17025 (DGR 473/2010). During the study period, the entire dog population entering in kennels was tested. The owned dogs were enrolled in specific research projects or sampled by veterinary practitioners for annual CanL screening, diagnosis or follow-up.

The dog identification was checked and microchip was implanted, if not found, by official and private veterinarians involved, as prescribed by the Italian Law (Legge 281/1991). Blood samples were collected for the following reasons: kennel admittance, annual screening, clinical suspicion, differential diagnosis and follow up after pharmacological treatment. The blood collection was performed in accordance with the European Legislation on Animal Welfare. Approval of an ethics committee was not required for sampling because a single blood collection sample is considered a routine procedure in domestic animals. The owned dogs were enrolled during routinely veterinary clinical activity. As inclusion criteria, only the dogs identified with a microchip were enrolled in this study. If more than one test was performed on the same animal in the study period, only the positive one or the one with the highest titre was considered. All dogs enrolled in this study had no history of previous vaccination against leishmaniasis.

Individual, environmental and spatial risk factors that might influence the dog exposure to *L. infantum* were studied. The following individual risk factors were considered: age ($<$2, 2-$<$5, 5-$<$7, $\geq$7 years), sex (male, female), size ($<$25 kg–small, $\geq$25 kg–large), breed (crossbreed, purebred), and coat length (long, medium, short). For purebred dogs, coat length and dog size were obtained by Ente Nazionale della Cinofilia Italiana [24]. The ownership (kennel dog, owned dog), attitude (pet, watchdog, hunting dog) and living with other dogs (yes, no) were the environmental risk factors considered. The following spatial variables were evaluated: distance from the sea (inland, coast: cut-off: 20 km from the sea [25]), altitude ($\leq$99, 100–499, $\geq$500 meters a.s.l.), and land cover (artificial, agricultural, forest/semi-natural areas). A Digital Elevation Model (20x20 m) was used for altitude (meters above sea level—a.s.l.) [22]. Corine Land Cover (CLC) [26] was used to determine the prevailing land cover in a 500 meters radius around each point, corresponding to the estimated flight range of phlebotomine sand flies [27]. Kennel and dogs' owners' address were georeferenced using ArcGIS 10.3. Data were recorded in a Microsoft Excel file.

## Laboratory analysis

The presence of anti-*Leishmania* IgG antibodies in dog sera was analysed by an in-house IFAT performed following the OIE recommendations, using a 1:80 cut-off. The test had a sensitivity (Se) of 96.0% and a specificity (Sp) of 98.0% [1]. Slides were prepared using promastigotes of *L. infantum* reference strain MHOM/TN/80/IPT1 (Istituto Superiore di Sanità) cultured at Istituto Zooprofilattico Sperimentale del Lazio e della Toscana "*M. Aleandri*". Reference negative and positive dog sera were used as negative and positive controls, while phosphate-buffered saline (PBS) was used as blank control. A 1:100 dilution of rabbit anti-dog IgG conjugated to Fluorescein Isothiocyanate (FITC) (Sigma-Aldrich, St Louis, MO, USA) was employed. Reading was performed by two independent observers using a fluorescence microscope. Samples were classified as positive if homogeneous cytoplasmatic, coarse speckled cytoplasmatic or membrane fluorescence of promastigotes was observed at a serum dilution $\geq$1:80; samples were considered as negative otherwise [28]. Positive were serially two-fold diluted to determine the end-point titre.

## Statistical analysis

Chi-square test was used to test the associations between possible risk factors and the presence of antibodies against *L.infantum*. When associations were significant, odds ratios (OR) and the

95% confidence interval (CI) were estimated by univariable logistic regressions. The variables with a bivariate p value ≤0.25 were included in a multivariable stepwise logistic model [29]. The likelihood of the final model was evaluated by Log likelihood chi square test (LRT). A p value <0.05 (two-tailed) was considered statistically significant.

The seroprevalence was calculated as the proportion between the number of seropositive dogs and the total number of dogs tested in the period 2010–2014. The apparent prevalence (AP; i.e. the proportion of a population that tests positive using a diagnostic method), and the 95% CI were calculated using a binomial distribution. The true prevalence (TP; estimated from AP adjusting for the diagnostic test Se and Sp), was calculated using the following Rogan and Gladen equation [30]: TP = (AP + Sp− 1)/(Se + Sp− 1).

TP and Blaker's 95% CI were calculated using Epitools epidemiological calculators [31]. Analyses were performed by SPSS Statistics Software v.25 (IBM SPSS Statistics).

## Results

A total of 13,292 dog sera were collected. The 91.2% (12,128) of sera were from kennel dogs, 5.0% (658) from owned dogs, while 3.8% (506 dogs) were unclassified. Crossbreed dogs were 81% (10,767/13,292) and purebred 19% (2,525/13,292). Dogs were aged between 1 month and 20 years. The 38.8% of dogs were under 2 years of age and the 54.1% were males. The age distribution was similar between male and female dogs (Fig 2). The five most frequent breeds were Maremma sheepdog (394) followed by German shepherd (363), English Setter (215), Pit Bull (193) and Rottweiler (130). Autochthonous breeds accounted for 26.2% (662), while the remaining 73.8% (1863) were non-autochthonous breeds.

### Risk factors

Most of the analysed risk factors were significantly associated with the presence of antibodies against *L. infantum* (chi square test, p<0.05) at the univariable analysis (Table 1). Age, ownership and attitude were confirmed as risk factors for seropositivity in the multivariate logistic regression model (LRT = 304.66) (Table 1).

The risk of being seropositive increased with the age resulting higher in dogs aged ≥7 years (OR 5.6, 95% CI: 4.1–7.7). Being a male resulted as a risk factor (OR 1.4, 95% CI: 1.2–1.6). A

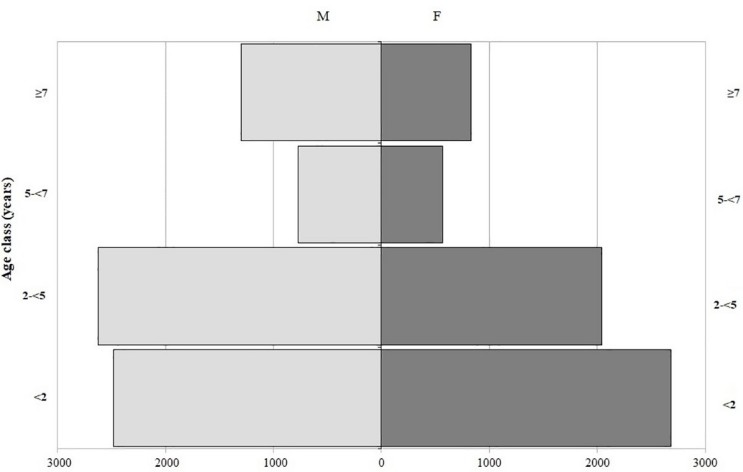

**Fig 2. Age pyramid of the dog population tested for *L.infantum*, in 2010–2014, in the Lazio region.**

**Table 1. Descriptive, univariable and multivariable logistic analyses of CanL seropositivity in 2010–2014, in the Lazio region.**

| Risk factors | Tested dogs (N) | Positive dogs (N) | Seroprevalence % (95% CI) | Univariable analysis OR (95% CI) | Multivariable analysis OR (95% CI) |
|---|---|---|---|---|---|
| **Individual variables** | | | | | |
| **Age** (years) | | | | | |
| <2 | 5161 | 134 | 2.6 (2.2–3.1) | - | - |
| 2-<5 | 4662 | 419 | 9.0 (8.2–9.8) | 3.7 (3.0–4.5) *** | 3.6 (2.6–4.8) *** |
| 5-<7 | 1341 | 209 | 15.6 (13.7–17.6) | 6.9 (5.5–8.7) *** | 5.4 (3.8–7.6) *** |
| ≥7 | 2128 | 336 | 15.8 (14.3–17.4) | 7.0 (5.7–8.7) *** | 5.6 (4.1–7.7) *** |
| **Sex** | | | | | |
| Female | 6116 | 421 | 6.9 (6.3–7.5) | - | |
| Male | 7176 | 677 | 9.4 (8.8–10.13) | 1.4 (1.2–1.6) *** | |
| **Size** | | | | | |
| <25 kg | 837 | 121 | 14.5 (12.1–7.0) | - | |
| ≥25 kg | 1689 | 191 | 11.3 (9.8–12.9) | 0.7 (0.6–1.0) * | |
| **Breed** | | | | | |
| Crossbreed | 10767 | 786 | 7.3 (6.8–7.8) | - | |
| Purebred | 2525 | 312 | 12.4 (10.1–13.7) | 1.8 (1.5–2.1) *** | |
| **Coat length** | | | | | |
| Long | 870 | 78 | 9.0 (7.1–10.1) | - | |
| Medium | 194 | 38 | 19.6 (14.2–5.9) | 2.5 (1.6–3.8) *** | |
| Short | 1043 | 156 | 5.0 (12.8–17.3) | 1.8 (1.3–2.4) *** | |
| **Environmental variables** | | | | | |
| **Ownership** | | | | | |
| Kennel | 12128 | 883 | 7.3 (6.8–7.8) | - | - |
| Owned | 658 | 489 | 74.3 (70.8–77.6) | 4.4 (3.6–5.3) *** | 2.8 (2.2–3.5) *** |
| **Attitude** | | | | | |
| Pet | 4337 | 420 | 9.7 (8.8–10.6) | - | - |
| Watchdog | 334 | 34 | 10.2 (7.1–13.9) | 1.1 (0.7–1.5) | 0.9 (0.6–1.4) |
| Hunting dog | 240 | 68 | 8.3 (22.7–34.5) | 3.7 (2.7–5.0) *** | 2.4 (1.7–3.4) *** |
| **Living with other dogs** | | | | | |
| No | 2987 | 209 | 7.0 (6.1–8.0) | - | |
| Yes | 10305 | 889 | 8.6 (8.1–9.2) | 1.3 (1.1–1.5) ** | |
| **Spatial variables** | | | | | |
| **Distance from the sea** | | | | | |
| Inland | 5689 | 475 | 8.3 (7.6–9.1) | | |
| Coast | 7342 | 610 | 8.3 (7.7–9.0) | | |
| **Altitude (m a.s.l.)** | | | | | |
| ≤99 | 5081 | 436 | 8.6 (7.8–9.38) | | |
| 100–499 | 7853 | 638 | 8.1 (7.5–8.7) | | |
| ≥500 | 97 | 11 | 11.3 (5.8–19.4) | | |
| **Land cover** | | | | | |
| Agricultural | 4532 | 381 | 8.4 (7.6–9.2) | - | |
| Artificial | 8383 | 674 | 8.0 (7.5–8.6) | 0.9 (0.8–1.1) | |
| Forest and semi natural | 377 | 43 | 11.4 (8.4–15.0) | 1.4 (1.0–2.0) * | |

CI: confidence interval; OR: odds ratio

*: p<0.05

**: p<0.01

***: p<0.001.

dog size ≥25 kg resulted as a protective factor (OR 0.7, 95% CI: 0.6–1.0). Purebred dogs were at higher risk to be seropositive (OR 1.8, 95% CI: 1.5–2.1). The comparison between autochthonous (662) and non-autochthonous breeds (1863) showed a higher risk of exposure in non-autochthonous breeds (OR = 1.5, 95% CI: 1.1–2.0, p<0.01). In detail, we observed a major risk in the other breeds respect to Maremma sheepdog (OR 1.9, 95% CI: 1.7–2.2, p<0.001). The coat length was available for 2,107 dogs out of 2,525 purebred dogs. Medium and short coat length were a risk factor for the exposure to *L. infantum* compared to breed characterized by long coat (OR 2.5 and OR 1.8 respectively). Owned dogs resulted at higher risk than kennel dogs (OR 2.8, 95% CI: 2.2–3.5). Hunting dogs were more at risk than pets (OR 2.4, 95% CI: 1.7–3.4). Living with other dogs was a significant risk factor (OR 1.3, 95% CI: 1.1–1.5). Regarding the spatial variable, distance from the sea and altitude did not affect the risk to develop antibodies against *L. infantum*, whereas the land cover was associated with higher risk for seropositivity. In detail, dogs living in areas with forest and semi natural land cover seemed to be at higher risk to develop antibodies against the parasite (OR 1.4, 95% CI: 1.0–2.0).

## Seroprevalence

In the whole population, the AP was 8.3% (95% CI: 7.8–8.7) while the TP was 6.7% (95% CI: 6.2–7.2). A TP of 7.3% (95% CI: 6.8–7.8) was estimated in kennel dogs, whereas owned dogs showed 74.3% (95% CI: 70.8–77.6). A bimodal relation was found between seropositivity and age: age classes 5–6 and ≥7 years showed higher prevalence (Fig 3). Serological results were stratified by end-point titre on annual basis (negative, 1/80, 1/160, 1/320, 1/640, 1/1280, 1/2560, 1/5120). The titre of 1/80 had a bimodal distribution with a first peak in January and a second peak in March, while the titre of 1/320 had one peak in February and one in April/May. The remaining titres, rather, did not show a specific pattern.

The seasonal profiles of the titres were reported in Fig 4. CanL seroprevalence in both kennel and owned dogs decreased during the study period (Fig 5).

## Discussion

### Risk factors

**Individual risk factors.** In the present study, the risk of being seropositive was higher starting from 2 years old, in agreement with several studies reporting that the risk of

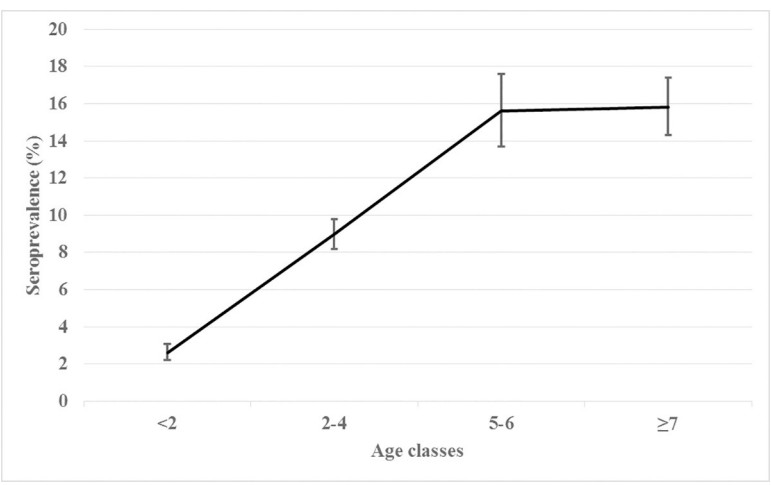

**Fig 3. Relation between CanL seroprevalence and age in 2010–2014, in the Lazio region.**

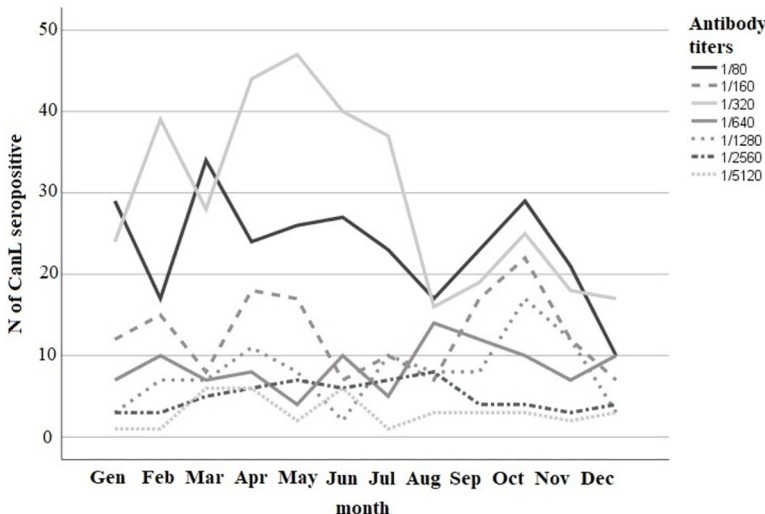

**Fig 4. Annual distribution of anti-*L. infantum* antibody titres in the dog population in 2010–2014, in the Lazio region.**

seropositivity increased with the dog age [14, 17, 32–35]. This evidence could be attributed to repeated exposure to *Leishmania* during the years. On the contrary, other authors reported a bimodal age distribution of seroprevalence with one peak appearing in the young dogs (1–2 years) and a second more evident peak among the older ones (7–8 years). This pattern was explained by the composition of the target population: dogs more sensitive to the disease (e.g. the Boxer) will develop it at an early age, whereas in more resistant dogs, latent infection will not be activated until they are older and their immune system decays, often in comorbidity with other diseases [36].

A significant difference in prevalence was found between sexes, with males showing higher seroprevalence. Other authors reported a higher seroprevalence in males [34, 36]. This result could be due to the roaming behaviour of male dogs in rural areas or the outdoor attitude of male guard dogs, as reported in a study carried out in the study area [21]. Conversely, other studies did not found significant difference between sexes [11, 37].

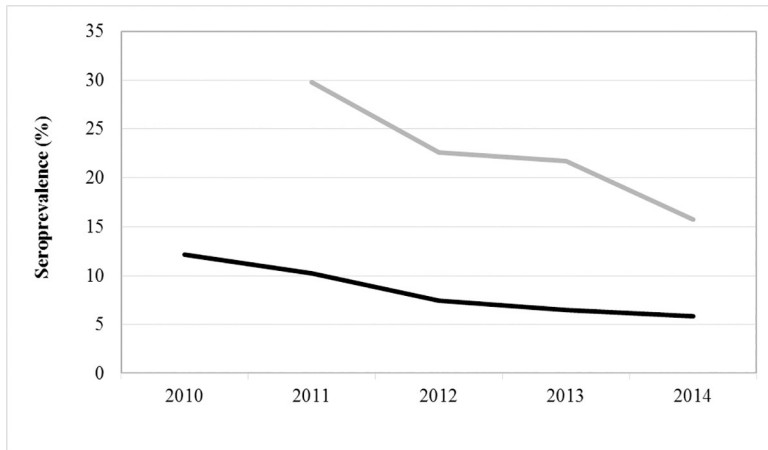

**Fig 5. CanL seroprevalence trend in kennel and owned dog population in 2010–2014, in the Lazio region.**

In this study, being large-size dogs (≥25 kg) was protective for CanL seropositivity as already reported [14]. A confounding effect due to the minor life expectancy in large-size dogs could explain this finding [38]. Purebred dogs showed a significant difference in seroprevalence, being a crossbreed resulted a protective factor, as previously reported [33, 39]. This result might be related to a certain level of resistance against the disease of crossbreed dogs. Conversely, some autochthonous purebred dogs, whose presence has been documented in the Mediterranean region since ancient times, have evolved under a strong selective pressure by *L. infantum*, developing a resistance to the disease. Among these breeds, the Ibizan hound, from the Balearic Islands, is certainly the most famous. This dog rarely develop clinical leishmaniasis [40, 41]. Nevertheless, studies are still rare in other autochthonous breeds and further researches are needed. It was demonstrated that resistant dogs mount a predominantly immune cellular response, protective against *Leishmania*. In contrast, susceptible dogs give a humoral, often exaggerated and not protective response [41]. In this study, the autochthonous Maremma sheepdog showed a low CanL seroprevalence. Similar values were also observed in another autochthonous Mediterranean breed, the Sardinian sheepdog (or Fonnese dog) [42]. This breed shared with the Maremma sheepdog, attitude and isolation conditions that preserved their relative purity over time [43].

We found an increasing risk of being seropositive for animals with short and medium coat length. Other authors found short hair dogs having a higher risk than the long hair ones [33, 44]. Indeed, this feature can facilitate the bite of phlebotomine sand flies who are known to feed on hairless area such as the margin of dog muzzle, which is always uncovered [33]. In addition, longer hair would reduce $CO_2$ emission and heat irradiation from the host's body thus being less attractive for vectors [45].

**Environmental risk factors.**  In this study, a significant difference in seroprevalence was found between kennel and owned dogs (7.3% vs 74.3%). This difference could be explained by a selection bias because owned dogs were more frequently tested for clinical suspicion or follow up after treatment, whereas kennel dogs were mainly sampled at kennel admittance. However, recent serosurveys on CanL reported higher risk for seropositivity in owned dogs compared to kennel [34, 35].

In this study, hunting dogs were more than twice fold at risk of being seropositive then pet dogs. The hunting activity imply prolonged outdoors staying with an increased exposure to phlebotomine sand flies and some studies indicated dogs living outdoors at higher risk than those with an indoor lifestyle [1, 11, 33, 36]. In addition, hunters are generally less prone to use topical insecticides than pet owners. Living with other dogs also appeared to be a risk factor for the exposure to *L. infantum* in sampled dog population in agreement with what previously observed [9, 46].

**Spatial risk factors.**  Until the late '80s, some typical spatial patterns of CanL were found in Italy such as distribution along the coast, in the Mediterranean belt and up to an average altitude of 500–600 m a. s. l. [7, 47]. Nowadays, the average increase in temperatures has likely shifted the classic climatic hot zones and the vectors distribution toward higher latitudes and altitudes, the latter often at larger distance from the coast [48]. In the Mediterranean basin, the distribution model of the main vectors of leishmaniasis has changed. A latitudinal shift towards North in Italy, in Spain and in France and towards higher altitudes in the southern Spain were documented [7, 49]. It is what most likely happened in the Lazio region too. A significant difference only in univariable analysis was found when prevailing land cover was 'forest' and 'semi natural' land cover. However, it is important to underline that micro-environmental (small-scale) factors on the distribution of several sand flies species are important in CanL epidemiology [50].

## Seroprevalence

An overall CanL TP of 6.7% was estimated. This value was lower than what observed in a previous study carried out in the same area in the period 2001–2004, reporting a raw prevalence around 25% [37]. This can be explained by the growing awareness of people on dog health and welfare that could have limited the spread of CanL during the years through a systematic anti-vectorial prophylaxis, early diagnosis and a proper treatment of infected individuals. Moreover, in the period covered by this study it is possible to notice that the seroprevalence followed a decreasing trend (Fig 5), as a result of an improved CanL management over the last years.

In addition, an increasing number of dogs identified by microchip in the Dog Registry and in Laboratory Informative System allowed a better traceability, excluding dog duplicates from the analysis that could have led to overestimate the prevalence in the past.

Another important topic is the seasonal pattern of the antibody titres. Antibody titre is an information that allows us to distinguish between infected but not sick dogs, with a tendency to low-medium titres, and sick dogs, with parasite dissemination and a tendency to high titres [16, 51]. The seasonal density of phlebotomine sand flies in the Mediterranean region has a bimodal trend encompassing the June-September period and the seroconversion occurs on average 5 months after the infection [52]. In the present study, the 1/80 antibody titre peaks occurred in March and October, 6 months after the probable exposure to the vector density peaks. These dogs probably represent recently exposed or infected animals [53]. The low antibody titres dynamics (Fig 4) is a partial, diachronic reflection of the vector's seasonal activity. A spatial approach, which takes this principle into account, could be useful to complement an annual CanL risk mapping activity, as suggested by other authors [54]. The vectorial activity is dynamic and a serological multi-titres surveillance in a population of georeferenced dogs could provide a more sensitive approach to the spatial representation of recent exposure areas as a clustering of samples with low serological titres.

## Kennels management

The stray dog management certainly represent a high cost in terms of dog maintenance and the potential risk of transmission of infectious diseases including leishmaniasis. Even if in a general decreasing trend framework of the leishmaniasis in Europe and Italy, kennels can play a role as hot spot of the disease, since they are often managed with scarce funding, the sanitary level is often low and dogs can be not properly treated with preventive treatments against ecto-parasites. For the same reason, infected dogs can be not properly treated when infected by *L. infantum*. In these cases, dogs may act as a source of infection for phlebotomine sand flies, in particular in crowded kennels and in endemic regions. Measures as a systematic use of anti-vectorial prophylaxis and the treatment of infected animals can improve the effectiveness of surveillance and control, improving the dogs welfare in kennels. Another measure could be housing the subjects at higher risk (older dogs, non-autochthonous breeds, etc.) into homogeneous risk classes. Avoiding the exposure of dogs to the bites of sand flies in the night is also a preventive measure to protect them from CanL.

Nevertheless, kennels offer a benefit in terms of active surveillance of leishmaniasis. A risk-based approach would be appropriate in the kennel management, through an active serological surveillance on dog (acting as sentinel) checked at kennel admittance and an entomological surveillance.

## Supporting information

**S1 Dataset.**
(XLS)

## Author Contributions

**Conceptualization:** Pasquale Rombolà, Gladia Macrì.

**Data curation:** Pasquale Rombolà, Gladia Macrì.

**Formal analysis:** Francesca Iacoponi.

**Software:** Pasquale Rombolà, Francesca Iacoponi.

**Supervision:** Pasquale Rombolà, Giulia Barlozzari, Andrea Carvelli, Manuela Scarpulla.

**Visualization:** Pasquale Rombolà.

**Writing – original draft:** Pasquale Rombolà, Francesca Iacoponi.

**Writing – review & editing:** Giulia Barlozzari, Andrea Carvelli, Manuela Scarpulla, Francesca Iacoponi, Gladia Macrì.

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
