## [Decision Letter · Decision Letter 0]

26 Oct 2020

PONE-D-20-21436

Risk factors for canine leishmaniasis in kennel and owned dogs in an endemic Mediterranean region

PLOS ONE

Dear Dr. Rombolà,

Thank you for submitting your manuscript to PLOS ONE. After careful consideration, we feel that it has merit but does not fully meet PLOS ONE’s publication criteria as it currently stands. Therefore, we invite you to submit a revised version of the manuscript that addresses the points raised during the review process.

We look forward to receiving your revised manuscript.

Kind regards,

Vyacheslav Yurchenko

Academic Editor

PLOS ONE

Additional Editor Comments:

The work has been reviewed by 3 specialists in the field and they all found the work interesting. I invite authors to address their concerns and submit the revised version.

Journal Requirements:

2. As part of your revisions, please address the following requests: (1) explain how you were able to access the sera samples collected by the veterinarians. As part of this explanation, please state whether you obtained any type of permission or authorization to obtain access to these samples. (2) Kindly provide details about consent that dog owners provided prior to blood collection by veterinarians.

3. We note that Figure 1 in your submission contain map images which may be copyrighted. All PLOS content is published under the Creative Commons Attribution License (CC BY 4.0), which means that the manuscript, images, and Supporting Information files will be freely available online, and any third party is permitted to access, download, copy, distribute, and use these materials in any way, even commercially, with proper attribution. For these reasons, we cannot publish previously copyrighted maps or satellite images created using proprietary data, such as Google software (Google Maps, Street View, and Earth). For more information, see our copyright guidelines: http://journals.plos.org/plosone/s/licenses-and-copyright.

3.1.    You may seek permission from the original copyright holder of Figure 1 to publish the content specifically under the CC BY 4.0 license. 

3.2.    If you are unable to obtain permission from the original copyright holder to publish these figures under the CC BY 4.0 license or if the copyright holder’s requirements are incompatible with the CC BY 4.0 license, please either i) remove the figure or ii) supply a replacement figure that complies with the CC BY 4.0 license. Please check copyright information on all replacement figures and update the figure caption with source information. If applicable, please specify in the figure caption text when a figure is similar but not identical to the original image and is therefore for illustrative purposes only.

Reviewers' comments:

Reviewer's Responses to Questions

**Comments to the Author**

1. Is the manuscript technically sound, and do the data support the conclusions?

Reviewer #1: Partly

Reviewer #2: Yes

Reviewer #3: Yes

2. Has the statistical analysis been performed appropriately and rigorously? 

Reviewer #1: N/A

Reviewer #2: Yes

Reviewer #3: Yes

3. Have the authors made all data underlying the findings in their manuscript fully available?

Reviewer #1: No

Reviewer #2: Yes

Reviewer #3: Yes

4. Is the manuscript presented in an intelligible fashion and written in standard English?

Reviewer #1: Yes

Reviewer #2: Yes

Reviewer #3: Yes

5. Review Comments to the Author

Reviewer #1: Comments - Manuscript Number: PONE-D-20-21436

Title: Risk factors for canine leishmaniasis in kennel and owned dogs in an endemic Mediterranean region

Authors: Pasquale Rombolà et al.

Reviewer Comments:

It is a generally well written manuscript, describing a seroprevalence study with risk factors assessment in a dog population of an endemic area for L. infantum zoonotic leishmaniasis in Italy. Although no striking novelty is brought, the study results’ contribution is valuable in epidemiological terms, mainly considering the number of dogs assessed and interesting results concerning the dynamics of the disease in the region. The work requires some writing edition and English revision.

Specific comments:

Abstract: Requires a better restructuration for more objectivity. Please clarify your objectives, describe your results accordingly and consistently provide your conclusions.

Line 24: Please describe the acronym IFAT the first time it appears.

Lines 32-32: For instance, the statement “This study confirmed the importance of the kennels as useful component for CanL active and passive surveillance.” is a bit vague and not exactly in accordance with the objectives of the study. If evaluating kennels was among the study’s objectives, it need to be clear.

Introduction: Please organize your ideas by separating statements and information on humans and animals’ leishmaniasis clearly, preferably in separate paragraphs. Please use indentation for paragraphs and avoid an excessive use of parentheses.

Line 47: Please replace “restricted” by a more appropriate word for the idea; perhaps “concentrated”.

Line 52: Please observe your statement: “According to some estimates, in southern Europe there are 700 new human cases of ZVL per year, about 200 of those cases occurring annually in Italy”. You are using references from 2010 and 2011, which is not consistent with the idea of “new cases”. Please observe the use of proper references and amend accordingly here and in the whole manuscript.

Lines 52-53: Please observe the phrase: “CanL causes a spectrum of clinical signs that varies greatly from asymptomatic/mild to a very severe disease.” CanL is the disease, it does not cause it. The clinical signs of an infected dog can vary from absent – in subclinical infections – to numerous and intense. The disease can vary from unapparent to mild or very severe. Amend accordingly.

Line 55: Please replace “are” by a more appropriate “can be”.

Lines 58-60: There are many differences between IFAT and ELISA for this purpose, depending on the antigen and many other aspects, and these varies considerably. So, the statement is a bit too generalized. Please amend describing the real limits of the used method – the IFAT, using proper references.

Line 64: Please replace the word “conclusive” by the more appropriate idea which is “strongly indicative” or something more like that. Only PCR and parasite isolation / observation are conclusive.

Lines 69-72: Please observe that these two phrases are quite antagonistic. Both ideas can be true. Please define your line of ideas clearly, because there seems to be an implicit objective in them. Make it explicit, which can bring value to your work. Did you intent to evaluate the kennel’s conditions or possible role in the epidemiology of leishmaniasis by L. infantum?

Materials and Methods: In general, it requires some more information. Was there any inclusion/exclusion criteria for study dogs?

Lines 84-85: please rewrite the phrase.

Line 89: “During ‘the implementation’ of the present study”… We are in 2020.

Line 116 and on: Please provide information on the clinical handling of the kenneled dogs; methods/criteria for examination and sample collection.

Line 123: Is this questionnaire directed to whom? It is published or disclosed in some publication? Is it validated? Otherwise, perhaps it is better to refer to a detailed anamnesis.

Lines 140-141: Please provide a little more rationale’s detail on AP and TP.

Lines 153-156: Please observe the policy of the Journal and the general legislation for the ethical approaches of research using animals. There was handling of animals in the kennels. They were examined and sampled.

Results: In general, the entire section needs some language edition and a more objective organization, at least by describing related results in accordance with the analytic method used, in separate paragraphs.

Line 189: Clinical signs in relation to serology should be described in a more complete way. “Etc” is not very precise. The literature is plenty of studies and data on the close relation between susceptibility, predominant immune response, parasite load, frequency and intensity of disease manifestation, infectiousness to the vector, and serology in dogs. Please amend. Is there a possibility of co-morbidities in the region?

Discussion: The section requires a better organization and separation of ideas by groups of related results in paragraphs. Please clearly separate literature citations from your own discussion and inputs based on your findings.

Lines 217-220: Please explore this result with more clarity and depth, preferably in a separate paragraph, taking into consideration inclusion and exclusion criteria. This should be clear in the abstract, accordingly.

Lines 229-231: Please elaborate this.

Line 235 and others: Leishmania should be italicized and written with an initial capital letter. Please observe it and amend it in the whole text.

Lines 241-247: These statements are a bit vague. How is the general attitude of owners towards male or female dogs? Are males preferred in the region, for hunting or guard, for example? Would they be more exposed to infection somehow?

Line 247 and on: Is the term “Conversely” appropriate here? The whole discussion about autochthonous breeds is very interesting and should be separate in another paragraph.

Lines 278-279: Please provide a reference for the statement “Nowadays, the average increase in temperatures has likely shift the classic climatic hot zones toward higher altitudes and larger distance from the coast”... in the studied region? Otherwise, the statement becomes speculative.

Lines 288-290: The statements: “For the same reason, infected dogs can be not properly treated when infected by L. infantum. In these cases, dogs may act as a source of infection for phlebotomine sand fly, in particular in overcrowded kennels and in endemic regions” – feel in disagreement with your own findings of a lower seroprevalence along time, for example, in comparison with a previous publication. The idea is valid, but please adopt a more rational insertion for this in the context of your own study and results.

Line 293: Is the general welfare of dogs in kennels in the region, or particularly in the kennels you have studied, compromised? Did you observe in them any aspect that should be described in this work and discussed? Shouldn’t it then be in the objectives, if relevant? Animal welfare is obviously relevant, but since this is as objective study, data on this aspect should be treated with the same objectivity as the other scientific data. Please amend.

Line 302: The conclusion should be in line with the objectives and results.

Figures and tables: All titles of figures and table should be more complete, with explanatory info enough to be understood without the reader having to go back to the text.

Reviewer #2: This is a very interesting paper focusing on the seroprevalence of CanL in the Mediterranean Region in Italy. Although the paper has local significance their findings are interesting especially if compared with other countries/regions. The paper is well-written and the analysis supported their findings and generated data. English is of high quality. I therefore recommend acceptance.

Reviewer #3: The manuscript contains a well conduced analysis of seroprevalence and risk factors to canine Leishmaniasis in Lazio region. The work is well designed, contains relevant information, and it is within the scope of the journal. I have some suggestions as described below:

Some suggestions:

M&M-

Line 104- Organization

Age classes are repeated in Line 123 and 126 .

Results

According with text, line 171-172 - “A bimodal relation was found between seropositivity and age: age classes 2-4 and >6 years showed higher prevalence (Fig 3).” However, in figure 3, the Y axis shows the number of seropositive dogs, this gives the false impression that there was a fall in the 5-6 age group. So, I suggest using % seropositive (seroprevalence).

6. PLOS authors have the option to publish the peer review history of their article (what does this mean?). If published, this will include your full peer review and any attached files.

Reviewer #1: No

Reviewer #2: No

Reviewer #3: No

---

## [Author Response · Author response to Decision Letter 0]

17 Dec 2020

In a separate file “rebuttal letter - response to reviewers” our answers to each point raised by the Academic Editor and the reviewers.

---

## [Editor Report · Decision Letter 1]

21 Dec 2020

Seroprevalence and risk factors associated with exposure to Leishmania infantum in dogs, in an endemic Mediterranean region

PONE-D-20-21436R1

Dear Dr. Rombolà,

We’re pleased to inform you that your manuscript has been judged scientifically suitable for publication and will be formally accepted for publication once it meets all outstanding technical requirements.

Kind regards,

Vyacheslav Yurchenko

Academic Editor

PLOS ONE
---

## [Editor Report · Acceptance letter]

23 Dec 2020

PONE-D-20-21436R1 

Seroprevalence and risk factors associated with exposure to *Leishmania infantum* in dogs, in an endemic Mediterranean region 

Dear Dr. Rombolà:

I'm pleased to inform you that your manuscript has been deemed suitable for publication in PLOS ONE. Congratulations! Your manuscript is now with our production department. 

Kind regards, 

on behalf of

Prof. Vyacheslav Yurchenko 

Academic Editor

PLOS ONE